# Internal consistency reliability, construct validity, and item response characteristics of the Kessler 6 scale among hospital nurses in Vietnam

Norito Kawakami[1]*, Thuy Thi Thu Tran[2], Kazuhiro Watanabe[1], Kotaro Imamura[1], Huong Thanh Nguyen[3], Natsu Sasaki[1], Kazuto Kuribayashi[4], Asuka Sakuraya[5], Quynh Thuy Nguyen[2], Nga Thi Nguyen[3], Thu Minh Bui[6], Giang Thi Huong Nguyen[6], Harry Minas[7], Akizumi Tsutsumi[8]

1 Department of Mental Health, Graduate School of Medicine, The University of Tokyo, Tokyo, Japan, 2 Department of Occupational Health and Safety, Faculty of Environmental and Occupational Health, Hanoi University of Public Health, Hanoi, Vietnam, 3 Faculty of Social Sciences—Behavior and Health Education, Hanoi University of Public Health, Hanoi, Vietnam, 4 Department of Psychiatric Nursing, Graduate School of Medicine, The University of Tokyo, Tokyo, Japan, 5 Department of Public Health, Tokyo Women's Medical University, Shinjuku-ku, Tokyo, Japan, 6 Nursing Office, Bach Mai Hospital, Hanoi, Vietnam, 7 Melbourne School of Population and Global Health, The University of Melbourne, Melbourne, Australia, 8 Department of Public Health, Kitasato University School of Medicine, Sagamihara, Kanagawa, Japan

* nkawakami@m.u-tokyo.ac.jp

## Abstract

The present study investigated the internal consistency reliability, construct validity, and item response characteristics of a newly developed Vietnamese version of the Kessler 6 (K6) scale among hospital nurses in Hanoi, Vietnam. The K6 was translated into the Vietnamese language following a standard procedure. A survey was conducted of nurses in a large general hospital in Hanoi, Vietnam, using a questionnaire including the Vietnamese K6, other scales (DASS21, health-related QOL, self-rated health, and psychosocial work environment), and questions about demographic variables. Internal consistency reliability (Cronbach's alpha coefficient) was calculated. A confirmatory factor analysis was conducted. Eleven hypotheses were tested (as Pearson's correlations with the K6) to assess the scale's construct validity. Item response theory (IRT) analysis was conducted to identify the item response characteristics. The Cronbach's alpha coefficient was 0.864. The explanatory and confirmatory factor analyses indicated a one-factor structure. Most hypotheses tested for construct validity were supported. IRT analysis indicated that response categories were located in order according to severity. K6 provided reliable information regarding higher levels of psychological distress. The findings suggest that the Vietnamese version of the K6 is a reliable and valid instrument to measure psychological distress among hospital nurses in Vietnam.

**Data Availability Statement:** All data files are available from the Figshare database (DOI: 10. 6084/m9.figshare.12005511.v2).

**Funding:** NK received a grant from Japan Agency for Medical Research and Development (AMED) under Grant Number JP18jk0110014 (https:// www.amed.go.jp/). The funder had no role in study design, data collection and analysis, decision to publish, or preparation of the manuscript.

**Competing interests:** NK reports grants from Fujitsu Ltd, SB AtWork Corp., and TAK Ltd, personal fees from Japan Dental Association, Junpukai Health Care Center, Occupational Health Foundation, Osaka Chamber of Commerce and Industry, SB AtWork Corp., and Sekisui Chemicals outside the submitted work. This does not alter our adherence to PLOS ONE policies on sharing data and materials The other authors declare that they have no competing interests.

## Introduction

The Kessler 6 (K6) scale is a six-item short measure of psychological distress and is also used as a screening instrument for common mental disorders in the community [1, 2]. The items were selected from a large item pool based on item response theory (IRT) analysis [1]. The interviewer-administered K6 showed a good screening performance for common mental disorders in the community, with the area under the curve (AUC) of receiver operating characteristic of 0.854, a sensitivity of 0.36 and a specificity of 0.96. The K6 was translated into 21 language families [3]. Because of its brevity, the scale has been used widely in mental health epidemiology. A cross-country validation study including data from 14 countries reported a substantial concordance between interviewer-administered K6 scores and diagnoses of common mental disorders (CMD) (the median AUC, 0.83) [4]. Country-specific validation studies have been published for the Arabic [5], English [6–8], Chinese [9], French [10], Japanese [11, 12], Korean [13], and Farsi (Persian) languages [14]. These studies in general reported good internal consistency reliability (0.69–0.92 in Cronbach's alpha) [5, 7, 9, 10, 12–15] and factor-based [9, 10] and other construct validity (correlations with other mental health measures) [5, 7, 8, 10, 13] across various populations (university students, community residents, employees, people living with HIV, and psychiatric patients), for both self-report and interviewer-administered versions. However, contrary to the initial assumption of uni-dimensionality [1, 2], some studies reported a multifactorial factor structure for the K6 [9, 10]. In addition, while a scoring approach based on unweighted item scores (i.e., 0-1-2-3-4 for the five response categories) has been used in most studies, the authors of the K6 argued that IRT-based weighted scoring could be a better and easy-to-use alternative [1]. In fact, an IRT analysis combining data from 14 countries suggested the use of a weighted scoring approach (e.g., 0-0-0-1-1) [4]. An IRT study reported that the psychometric properties of the K6 may vary depending on ethnicity in the United States [16]. Psychometric properties of the K6 should be systematically investigated including confirmatory factor analysis and IRT analysis for specific country and ethnic populations [17].

There is a growing concern for mental health in low- and middle-income countries (LMICs) [18]. One of the proposed strategies to scale up mental health service in these countries is to integrate mental health services into workplaces [19]. In LMICs, healthcare professionals face increasing stress and burnout at work [20] because of a shortage of healthcare professionals [21, 22], population aging, and demands from service users (i.e., patients and families) [23–25]. This is particularly the case in South-East Asia [21]. Increasing stress at work has been reported among nurses in Vietnam [26, 27], Thailand [28], and Myanmar [29]. High levels of stress at work and poor mental health of healthcare professionals may diminish the quality of health care services in these countries [30]. It is important to manage work-related stress and to improve mental health among nurses in Southeast Asian countries.

The present study investigated psychometric properties, i.e., internal consistency reliability, construct validity, and item response characteristics of the K6 scale translated in Vietnamese in a sample of nurses working in a large general hospital in Hanoi, Vietnam. This is the first validation study of a Vietnamese version of the K6. While the study used a sample from a specific occupational group the findings may be useful in exploring the cross-cultural validity of the K6 scale in Vietnam. This is also the first study in which IRT analysis was conducted on data from the K6 in a LMIC since a previous cross-national study used a combined data set from high-income countries and LMICs to conduct an IRT analysis [4].

## Methods

### Participants

A questionnaire survey was conducted as the baseline survey for a stress management intervention study in a large general hospital in Hanoi, Vietnam [31], from August 21 to September 10, 2018. Out of all registered nurses (n = 1,269) in this hospital, a total of 1,258, excluding those who were about to retire (n = 11), were invited to complete a questionnaire and return it in an envelope to collection boxes. The returned questionnaires were collected by staff at Hanoi University of Public Health and entered into a database.

The aim and procedure of the study was reviewed and approved by the Research Ethics Committee of the Graduate School of Medicine/Faculty of Medicine, The University of Tokyo (no. 11991-(1)) and the Ethical Review Board for Biomedical Research, Hanoi University of Public Health (no 346/2018/YTCC-HD3).

### Measures

**K6 scale.** The K6 scale is a 6-item self-report measure of psychological distress, with a 5-point response option (4 = All of the time, 3 = Most of the time, 2 = Some of the time, 1 = A little of the time, and 0 = None of the time) [1]. The possible range of the total score is 0 to 24, with higher scores indicating greater psychological distress. The translation of the K6 into the Vietnamese language was conducted following the standard procedure of translation and cultural adaptation of patient reported health outcomes [32]. After obtaining permission to translate it into Vietnamese from the lead author of the research team that developed the K6 (Prof Ronald C Kessler, Harvard Medical School, United States of America), the original English version was translated into Vietnamese independently by two translators who were fluent in English and were knowledgeable about mental health symptomatology. These translations were integrated into a preliminary translation. This translation was reviewed by collaborators at the Hanoi University of Public Health, and a brief cognitive interview was conducted of a group of nurses (n = 30) to ask their opinions on draft items to know if the wordings were meaningful and relevant, followed by necessary amendments. This amended version was back-translated into English by an independent translator, and reviewed by staff of the research team at Harvard Medical School, as well as by the authors. Further careful revisions were made to make sure some of the K6 items were appropriately translated: "hopeless" in item #2; "restless or fidgety" in item #3; "depressed" in item #4. The final draft was pre-tested in a pilot study of 150 nurses in the same hospital in June 2018. A psychometric analysis showed a problem in this draft: item #5 ". . .that everything was an effort?" did not correlate with other items. The item was further revised based on a discussion among the collaborators. The final version of the Vietnamese version of K6 was posted on the website of the Harvard Medical School: https://www.hcp.med.harvard.edu/ncs/k6_scales.php

**Other scales for testing the construct validity.** *Depression, anxiety and stress.* The 21-item version of the DASS scale, which measures symptoms of depression, anxiety, and stress in community settings [33, 34], is comprised of the three corresponding subscales, each with seven items. Items are scored on a 4-point scale ranging from 0 (*did not apply to me at all*) to 3 (*applied to me very much, or most of the time*). Each subscale score ranges from 0 to 21. Reliability and validity of the Vietnamese version of the DASS21 scale have previously been reported [35].

*Health-related quality of life.* The EQ-5D-5L is a widely used indicator of health-related quality of life (HR-QOL) [36]. The EQ-5D-5L consists of five items covering mobility, self-care, usual activities, pain/discomfort and anxiety/depression, each of which is rated on a five

category scale from no problems (1) to extreme problems (5). Reliability and validity of the EQ-5D-5L are well established [36]. A Vietnamese version of DASS21 has been developed and tested, and has acceptable reliability and validity [37]. We used the standard value set for Vietnam to calculate participants' HR-QOL scores [38].

*Psychosocial work environment.* The Job Content Questionnaire (JCQ) was used to assess the psychosocial work environment [39]. It includes four scales: a five-item psychological demand scale, a nine-item decision latitude scale, a four-item supervisor support scale, and a four-item coworker support scale. The items are scored on a 4-point Likert scale, ranging from 1 (*strongly disagree*) to 4 (*strongly agree*). The reliability and validity of the Vietnamese version of the JCQ has been found to be acceptable, while Cronbach's alpha coefficients were lower for job demands and job control (0.446–0.499) than those for supervisor and coworker support (0.856–0.868) [40].

*Self-rated health.* Participants was asked to rate their health status on a 0–100 scale, in which 0 indicates the worst health status and 100 indicates the best health status. This was part of the EQ-5D-5L scale [36, 37], but not used in the calculation of the HR-QOL score.

**Demographic variables.** Demographic variables were included in the questionnaire. Respondents were asked to indicate their sex (male or female), and their birth year. Personal income per month was assessed by a single-item question, "What is your current monthly income (including salary and any other sources of income)?", with a 3-point response scale: $\leq$ 5 million VND (220 USD) (1), 5–10 million VND (220–432 USD) (2), and $\geq$ 10 million (432 USD) (3). Respondents were also asked whether they were currently married, never married, or divorced/separated, and about the nature of their labor contract: fixed-term contract for less than one year, fixed-term contract for more than one year, unspecified-term contract, permanent contract, or "other."

## Statistical analysis

Distribution of the K6 scores was examined separately for men and women. The average and standard deviation (SD), maximum and minimum values of K6 scores were calculated. The proportions of participants who had a score of 13 or higher [2] were calculated for men and women. Internal consistency reliability was calculated by using the Cronbach's alpha coefficient. Then a confirmatory factor analysis was conducted to assess the fit of the data to a one-factor structure. The goodness of fit index (GFI), adjusted goodness of fit index (AGFI), and comparative fit index (CFI) greater than 0.95 and the root mean square error of approximation (RMSEA) smaller than 0.06 to 0.08 were considered to be indicators of acceptable fit [41] For a comparison with previous studies, an exploratory factor analysis was conducted with maximum likelihood extraction, in order to assess the number of factors using a scree plot and proportion of variance explained by the first factor.

Construct validity was assessed by using Pearson's correlations (rs), with the following 11 hypotheses: DASS21 scores of (1) depression, (2) anxiety, and (3) stress moderately or strongly and positively (rs>0.4) with K6 score [5, 10, 13]; (4) HR-QOL and (5) self-rated health status moderately and negatively (rs = -0.4 to -0.7) correlated with K6 scores [42]; (6) job demands weakly or moderately positively (r = 0.2 to 0.4), and (7) job control, (8) supervisor support, and (9) coworker support weakly or moderately negatively (rs = -0.2 to -0.4) correlated with K6 scores [43]; socioeconomic position, such as (10) higher education and (11) personal income weakly and negatively (rs = -0.1 to -0.2) correlated with K6 scores [44].

A screening performance of K6 for identifying a CMD (i.e., depressive disorders and anxiety disorders) was also examined. Cases with CMD were defined as those who had 34 or greater scores of the total DASS21, that was the best cut-off score found in a previous study in

Vietnam [35]. An area under the curve (AUC) of a receiver operative curve (ROC) and the 95% CIs were calculated predicting these cases with CMD on the sore of K6. An optimum cut-off score was determined based on the best Youden index (a sum of sensitivity and specificity), and the sensitivity, specificity, and positive likelihood ratios were calculated with their standard errors [45].

The IRT analysis was conducted using the graded response model. Threshold parameters for response categories and slope parameter of each item were estimated. The test information curve and item information curves were presented in a graphical way.

Most of these analyses were conducted using IBM SPSS version 22. The confirmatory factor analysis and IRT were conducted using PROC CALIS and PROC IRT, respectively, from the SAS software package version 9.4.

## Results

### Respondent's characteristics

A total of 949 (75.6% of the target population) respondents returned the questionnaires. One respondent had a missing response on one of the K6 items, and was excluded from the analysis. Participants were mostly women, married, and younger than 40 years old, with age range between 22 and 58 years (Table 1). Most were with monthly personal income of more than 5 million VND (equivalent to 220 USD) About half had completed college or higher education. About half had a permanent contract, while about one fourth were employed with a fixed-term contract for one year.

### Distribution of the scale scores

The K6 scores showed a skewed distribution for both men and women (S1 Appendix). The average K6 scores of women and men did not significantly differ (Table 2). The proportion of respondents with a K6 score of 13 or greater was 2.4%, and did not significantly differ between men and women. Average item scores were higher for item #1 compared to those for the other items (Table 3). For items #4 and #5, no respondents endorsed the extreme category "all of the time").

### Internal consistency reliability

The Cronbach's alpha coefficient for the total K6 score was 0.864. Most items substantially contributed to the Cronbach's alpha coefficient, while item #1 did not (Table 3).

### Factor-based validity

The confirmatory factor analysis assuming a one factor structure indicated a moderate fit of the one factor model: GFI = 0.957; AGFI = 0.901; RMSEA = 0.115 (90% confidence intervals, 0.098 to 0.134); CFI = 0.954. Standardized coefficients (error variance) for items 1 to 6 were 0.582 (0.662), 0.764 (0.416), 0.726 (0.472), 0.774 (0.400), 0.794 (0.369), and 0.695 (0.517), respectively. In the explanatory factor analysis, initial eigenvalues (% of variance explained) for factors 1 to 6 were 3.619 (60.3), 0,750 (12.5), 0.472 (7.9), 0.449 (7.5), 0.399 (5.2), and 0.311, respectively, yielding one single factor.

### Construct validity

DASS21 scores for depression, anxiety, and stress correlated moderately and positively with K6 score (rs = 0.513 to 0.544) (Table 4). HR-QOL and self-rated health correlated negatively with K6 score (rs = -0.399 and -0.355, respectively), while the correlation coefficients were

**Table 1. Demographic characteristics of the study participants, hospital nurses in Vietnam (n = 948).**

| Variables | n | % |
|---|---|---|
| Sex | | |
| Men | 143 | 15.1 |
| Women | 805 | 84.9 |
| Age in years | | |
| 18–29 | 348 | 36.7 |
| 30–39 | 428 | 45.1 |
| 40+ | 172 | 18.1 |
| Education | | |
| Vocation school | 442 | 46.6 |
| Colleges | 137 | 14.5 |
| University undergraduate | 348 | 36.7 |
| Postgraduate | 17 | 1.8 |
| Unknown† | 4 | 0.4 |
| Marital status | | |
| Single | 136 | 14.3 |
| Married | 793 | 83.6 |
| Divorced/widowed | 16 | 1.7 |
| Unknown† | 3 | 0.3 |
| Personal income per month in VDN | | |
| $\leq$ 5 million (220 USD) | 91 | 9.6 |
| 5–10 million (220–432 USD) | 563 | 59.4 |
| $\geq$ 10 million (432 USD) | 284 | 30.0 |
| Unknown† | 10 | 1.1 |
| Employment contract | | |
| Fixed-term, <1 year | 219 | 23.1 |
| Fixed term, > 1 year | 29 | 3.1 |
| No fixed-term | 194 | 20.5 |
| Permanent | 504 | 53.2 |
| Others | 2 | 0.2 |

† Treated as missing in the analysis.

slightly smaller than 0.4. Job demands weakly and positively correlated with K6 score (r = 0.292). Job control, supervisor support and coworker support weakly and negatively correlated with K6 score (rs = -0.147, -0.204, and -0.160, respectively). Personal income was only marginally significantly associated with K6 (r = -0.06, p = 0.066).

Among the participants, 22 cases were identified as having CMD (i.e., 34 or greater scores of the total DASS21). The AUC predicting CMD on K6 score was 0.919 (95% CIs, 0.855–0.984). The best cut-off score for K6 was identified as 9+, with 129 (12.4%) of the participants having the scores equal to or greater than this cut-off. The sensitivity and specificity were 0.864 (SE, 0.073) and 0.876 (SE, 0.031), respectively, with the positive likelihood ratio of 6.96 (SE, 0.121).

## Item-response theory analysis

For all the items, the threshold parameters almost linearly increased from mild to severe response categories (Table 5). For items #2–6, the threshold parameters for most response categories of each item were greater than 0. On the other hand, for item #1, the first two threshold

**Table 2. Average, standard deviation (SD), median, minimum and maximum values of K6 scores in a sample of hospital nurses in Vietnam.**

| Sex* | N | Average | SD | Median | Minimum | Maximum |
|---|---|---|---|---|---|---|
| Men | 143 | 4.14 | 3.39 | 3.4 | 0 | 17 |
| Women | 805 | 4.64 | 3.51 | 3.9 | 0 | 20 |
| Total | 948 | 4.57 | 3.50 | 3.8 | 0 | 20 |

* No significant difference in the average score (t-test, p = 0.114) between men and women.

parameters were less than 0. The slope parameters were similar for items #2–6 (1.512–1.770), while the parameter was smaller for item #1 (0.909). The test information curves indicated that the K6 provided more information when the trait was greater than 0 (i.e., when people had higher psychological distress than the average) (Fig 1). Item information curves indicated that most of the items provided good information where the trait was greater than 0 (Fig 2). Item #1 had markedly lower information across the latent trait continuum than the other items. However, item #1 provided better information than other items where the trait was less than -0.5.

## Discussion

The Vietnamese self-report version of the K6 showed high level of internal consistency reliability. The uni-dimensional structure of the K6 was supported by the confirmatory factor analyses of the items. Most hypotheses tested for construct validity were supported. Item response analysis indicated that response categories were located in order according to severity. The K6 provided information for detecting higher psychological distress as a whole, but item #1 did not. The study suggests that the Vietnamese version of the K6 is a reliable and valid instrument to measure psychological distress.

The confirmatory factor analysis indicated a good fit for the one-factor structure, with acceptable levels of most goodness of fit indicators, while RMSEA was not within an ideal range (<0.6–0.8) but close. The findings of the confirmatory factor analysis suggest that the Vietnamese K6 has a one-factor structure. The first factor had an eigenvalue of 3.6 and explained 60.3% of the total variance in the explanatory factor analysis. The one-factor structure was consistently observed across 14 countries including LMICs (Brazil, China, Colombia, India, Mexico, Nigeria, Ukraine) [4] with similar eigenvalues for the first factor. The one-factor structure seems applicable to most countries, including LMICs. However previous studies

**Table 3. Average and standard deviation (SD) of item scores, item-total correlation, and Cronbach' alpha coefficient when an item was deleted for K6 in a sample of hospital nurses in Vietnam (n = 948).**

| Items | Item scores | | | Item-total correlation (corrected) | Cronbach' alpha coefficient when the item was deleted† |
|---|---|---|---|---|---|
| | Average | SD | Range | | |
| 1. . . .nervous | 1.66 | 0.83 | 0–4 | 0.543 | 0.864 |
| 2. . . .hopeless | 0.61 | 0.78 | 0–4 | 0.718 | 0.830 |
| 3. . . .restless or fidgety | 0.68 | 0.75 | 0–4 | 0.678 | 0.838 |
| 4. . . .so depressed that nothing could cheer you up | 0.77 | 0.81 | 0–3 | 0.703 | 0.833 |
| 5. . . .that everything was an effort | 0.53 | 0.71 | 0–3 | 0.720 | 0.831 |
| 6. . . .worthless | 0.31 | 0.63 | 0–4 | 0.615 | 0.850 |

† Cronbach's alpha coefficient for the total K6 score was 0.864.

**Table 4. Eleven hypotheses tested for the construct validity of K6 among hospital nurses in Vietnam: Pearson's correlation coefficients (rs) with K6 scores.**

| Variables | n† | Average | SD | rs | p |
|---|---|---|---|---|---|
| DASS depression score (0–21) | 932 | 3.0 | 2.9 | 0.513 | <0.001 |
| DASS anxiety score (0–21) | 935 | 3.9 | 3.1 | 0.539 | <0.001 |
| DASS stress score (0–21) | 935 | 5.6 | 3.6 | 0.544 | <0.001 |
| HR-QOL score (0–1) | 948 | 0.9 | 0.1 | -0.399 | <0.001 |
| Self-rated health (0–100) | 946 | 85.7 | 11.6 | -0.355 | <0.001 |
| JCQ job demands score (12–48) | 933 | 31.5 | 4.4 | 0.279 | <0.001 |
| JCQ job control score (24–96) | 931 | 81.1 | 6.4 | -0.147 | <0.001 |
| JCQ supervisor support score (4–16) | 943 | 12.0 | 1.9 | -0.204 | <0.001 |
| JCQ coworker support score (4–16) | 946 | 12.2 | 1.5 | -0.160 | <0.001 |
| Education (1–4) ‡ | 944 | 1.9 | 1.0 | -0.083 | <0.001 |
| Personal income (1–3) § | 938 | 2.2 | 0.6 | -0.060 | 0.066 |

† The number of participants varied due to missing responses on each variable.

‡ Coded as 1 = vocation school, 2 = colleges, 3 = university, 4 = postgraduate, excluding missing responses

§ Coded as 1 = ≤ 5 million (220 USD), 2 = 5–10 million (220–432 USD), 3 = ≥ 10 million (432 USD) per month, excluding missing responses.

in France and China reported a two-factor structure consisting of items related to depression and anxiety [9, 10, 15]. The factor structure of the K6 may still vary depending on language, ethnicity, and culture.

For construct validity, all of the 11 hypotheses tested were supported by the data. DASS21 scales for depression, anxiety, and stress all correlated moderately with the K6, which is concordant with previous findings of moderate to strong correlations with other depression scales [5, 10, 13]. Consistent with previous studies [42], HR-QOL and self-rated health also moderately correlated with K6 score. The correlations were slightly below 0.4. Job demands, job control, supervisor support, and coworker support correlated weakly ($|rs|$ = 0.15–0.28) in expected directions [43]. The modest associations between job demands and job control with K6 scores may be attributable to lower reliability of these scales [40]. Educational attainment and personal income correlated very weakly ($|rs|$<0.1) with the K6. These findings are in line with previous studies on socioeconomic factors associated with depression [44]. The small correlation for educational attainment may be a result of the fact that all participants had the same occupation; the small correlation for personal income may be attributable to some participants having income from other family members. The present study suggests an adequate level of construct validity of the Vietnamese version of the K6.

A previous IRT analysis using data from 14 countries found the two most severe response categories were useful for discriminating degrees of psychological distress, and discussed the use of a weighted scoring approach (e.g., 0-0-0-1-1) [4]. However, in the present study, the threshold parameters almost linearly increased according to the severity of response categories. The present study rather supports the 0-1-2-3-4 scoring for the Vietnamese K6. The test information curve was better when the latent trait was greater than 0, suggesting that the Vietnamese version of the K6 measure high levels of psychological distress more accurately. This version of the K6 may be useful for detecting poor mental health among hospital nurses in Vietnam, although how K6 score relates to probability of presence of diagnosable psychiatric disorder, and what is the most appropriate cutoff score, in Vietnam is currently unknown.

Internal consistency reliability measured by Cronbach's alpha coefficient was high enough in this sample to be comparable with previous reports [5, 9, 10, 12–15]. However, item #1 did not contribute much to the internal consistency reliability. The item information curves from

**Table 5. Response category characteristic curves for each item of K6 among hospital nurses in Vietnam.**

| Item | Parameter | Estimate | Standard error | Pr > \|t\| |
|------|-----------|----------|----------------|-----------|
| k1 | Threshold 1 | -2.162 | 0.129 | < .0001 |
| | Threshold 2 | -0.291 | 0.062 | < .0001 |
| | Threshold 3 | 1.564 | 0.101 | < .0001 |
| | Threshold 4 | 3.578 | 0.252 | < .0001 |
| | *Slope* | *0.909* | *0.060* | *< .0001* |
| k2 | Threshold 1 | 0.164 | 0.048 | 0.0004 |
| | Threshold 2 | 1.285 | 0.069 | < .0001 |
| | Threshold 3 | 2.420 | 0.128 | < .0001 |
| | Threshold 4 | 3.331 | 0.267 | < .0001 |
| | *Slope* | *1.536* | *0.104* | *< .0001* |
| k3 | Threshold 1 | -0.058 | 0.051 | 0.1256 |
| | Threshold 2 | 1.320 | 0.075 | < .0001 |
| | Threshold 3 | 2.987 | 0.183 | < .0001 |
| | Threshold 4 | 3.564 | 0.285 | < .0001 |
| | *Slope* | *1.320* | *0.087* | *< .0001* |
| k4† | Threshold 1 | -0.185 | 0.049 | < .0001 |
| | Threshold 2 | 1.096 | 0.064 | < .0001 |
| | Threshold 3 | 2.267 | 0.116 | < .0001 |
| | *Slope* | *1.512* | *0.100* | *< .0001* |
| k5† | Threshold 1 | 0.259 | 0.046 | < .0001 |
| | Threshold 2 | 1.384 | 0.071 | < .0001 |
| | Threshold 3 | 2.805 | 0.166 | < .0001 |
| | *Slope* | *1.770* | *0.133* | *< .0001* |
| k6 | Threshold 1 | 0.853 | 0.058 | < .0001 |
| | Threshold 2 | 1.841 | 0.098 | < .0001 |
| | Threshold 3 | 2.712 | 0.162 | < .0001 |
| | Threshold 4 | 3.424 | 0.276 | < .0001 |
| | *Slope* | *1.513* | *0.121* | *< .0001* |

† No respondent endorsed the highest response category for these items.

the IRT analysis indicated that item #1 provided poorer information than the other items. This is partly because the item #1 that originally asks "nervousness" was translated by using a Vietnamese word "stress", that is popular wording used frequently in daily life. The item #1 can be dropped from the Vietnamese version of the K6, still keeping the current level of internal consistency reliability and the scale information. However, it might be good to have an item with such lower difficulty in the beginning of the scale, to make it easy for respondents to start responding to the scale. Potential benefits of keeping and dropping this item should be investigated further, e.g., by conducting a randomized controlled trial comparing psychometrics of the two versions.

The distribution of scores of the Vietnamese version of the K6 was skewed to the right. The pattern is similar to an exponential distribution of K6 scores found in a study in the United States [46]. However, there was a peak at scores of 1–3 in this sample, not at the score of 0 as in the US sample [46]. This may be because the item response function of item #1 for those response categories had lower difficulty parameters. It is hard to compare average scores of K6 in this sample (4.57) with other reports, since some reports did not indicate the average score, and available average scores varied a lot (e.g., 3.6 in a community sample in Japan [12]; 5.9 in a

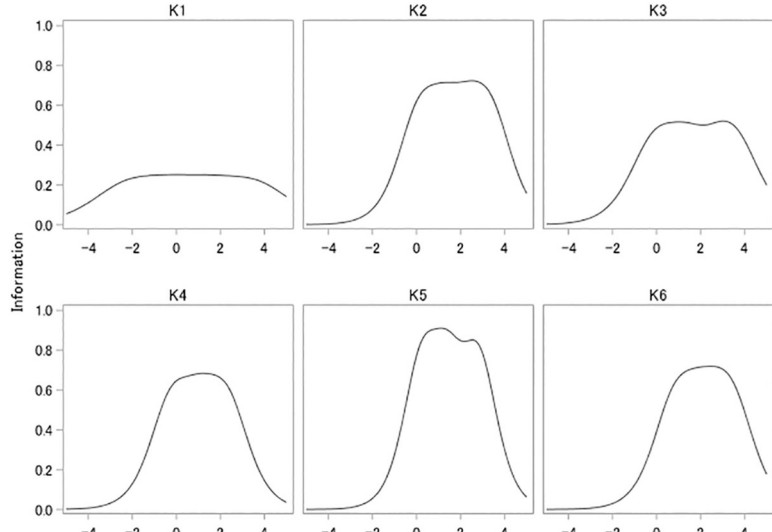

**Fig 1. Test information curve of K6 among hospital nurses in Vietnam.** The vertical axis indicates the latent trait of psychological distress.

community sample in the United States [1]; 10.0 among older people in Korea [13]; and 12.9 among Palestinian social workers [5]. The screening performance of K6 score predicting CMD defined by using DASS21 was high: the AUC was 0.919, with the sensitivity of 0.864 and the specificity of 0.876 for the best cut off score of 9+. The finding is similar to one from a previous study in Japan that the stratum-specific likelihood ratio became higher (>11) for the cut off scores of 9+ on K6 [47]. However, the screening performance and the best cut off score of the Vietnamese version of the K6 should be assessed by comparing people who have or have not been previously diagnosed with mental disorders.

A common use of the K6 (and K10) is in large-scale population mental health surveys. Despite some reservations about varying sensitivity of K6 for different disorders [48] it is a

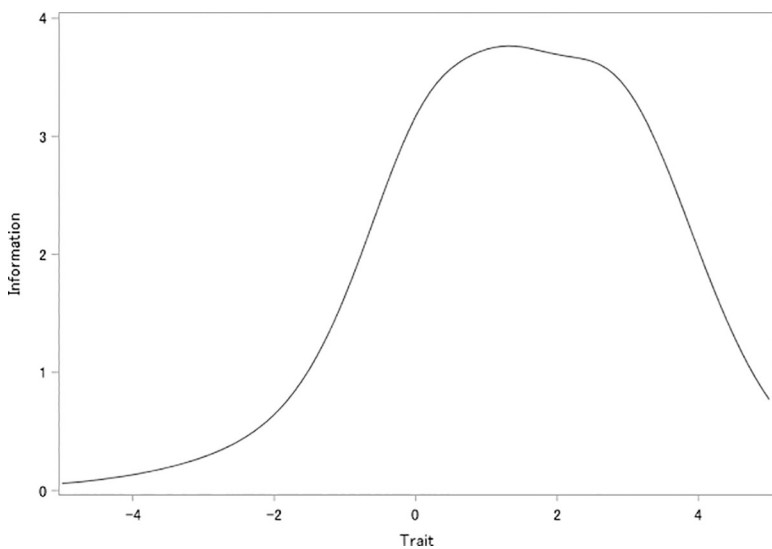

**Fig 2. Item information curve for each item of K6 among hospital nurses in Vietnam.** The vertical axis indicates the latent trait of psychological distress.

strong candidate for such use in Vietnam, where a population representative national mental health survey to estimate prevalence of common mental disorders has not yet been conducted. The absence of reliable estimates of population prevalence of mental disorders is a significant problem for mental health policy-making and advocacy. The use of the K6 in such national surveys would require studies to establish the criterion validity of the K6 in Vietnam.

The present study has several limitations. First, we aimed to validate the K6 scale to screen psychological distress among nurses, a special occupational group under high stress, in Vietnam. Nurses are usually more educated than general workers, and also familiar with questions of psychological symptoms included in K6. The present findings may not be generalized to the general population or working population in Vietnam. The reliability and validity of the present scale should be further tested in these populations. Second, as we noted earlier, we did not investigate the scale's screening performance for clinically diagnosed mental disorders. Third, the study did not examine test-retest reliability or indicators for interpretability (e.g., minimal important difference or minimally important change) which might be useful in applying the K6 in practice. Finally, 150 nurses who responded to a pilot study were also invited to the main validation study; most of them probably actually participated. Because the pilot study was anonymous, it was impossible to conduct an analysis excluding those who participated in the two surveys. Repeated surveys have been reported to result in attenuated responses to anxiety symptoms of respondents [49], which may affect the finding of the present study. Despite these limitations, the study clarifies the psychometric properties of the newly developed Vietnamese self-report version of the K6, including item response characteristics first examined in a LMIC.

## Conclusions

The present study found an acceptable level of internal consistency reliability, and supported a unidimensional factor structure and other construct validity of Vietnamese self-report version of the K6. Item response analysis indicated that the K6 provided information for detecting higher psychological distress as a whole. It is suggested that the Vietnamese version of the K6 is a reliable and valid instrument to measure psychological distress at least among nurses in Vietnam.

## Supporting information

**S1 Appendix. Distribution of K6 score among males and females in a sample of hospital nurses in Vietnam.**
(TIF)

## Acknowledgments

The authors acknowledged the support and contribution of the Director Board and the Nursing Office in Bach Mai hospital to the implementation of this study. We thank all nurses who participated in and provided their information for the study and highly appreciated the contribution of a research team from Hanoi University of Public health, an important partner to make this study possible.

## Author Contributions

**Conceptualization:** Norito Kawakami.

**Data curation:** Thuy Thi Thu Tran, Kotaro Imamura, Natsu Sasaki, Quynh Thuy Nguyen, Nga Thi Nguyen, Thu Minh Bui, Giang Thi Huong Nguyen.

**Formal analysis:** Norito Kawakami.

**Funding acquisition:** Norito Kawakami.

**Investigation:** Natsu Sasaki, Kazuto Kuribayashi, Asuka Sakuraya.

**Methodology:** Kazuhiro Watanabe, Huong Thanh Nguyen.

**Project administration:** Norito Kawakami, Thuy Thi Thu Tran, Kotaro Imamura, Huong Thanh Nguyen, Natsu Sasaki, Quynh Thuy Nguyen, Thu Minh Bui, Giang Thi Huong Nguyen.

**Resources:** Huong Thanh Nguyen, Harry Minas, Akizumi Tsutsumi.

**Supervision:** Thuy Thi Thu Tran, Quynh Thuy Nguyen, Nga Thi Nguyen, Giang Thi Huong Nguyen.

**Validation:** Thu Minh Bui.

**Writing – original draft:** Norito Kawakami, Natsu Sasaki, Kazuto Kuribayashi, Asuka Sakuraya, Quynh Thuy Nguyen, Thu Minh Bui, Harry Minas, Akizumi Tsutsumi.

**Writing – review & editing:** Norito Kawakami, Thuy Thi Thu Tran, Kazuhiro Watanabe, Kotaro Imamura, Huong Thanh Nguyen, Nga Thi Nguyen, Giang Thi Huong Nguyen.

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
