## [Decision Letter · Decision Letter 0]

10 Mar 2020

PONE-D-19-34231

Internal consistency reliability, construct validity, and item response characteristics of the Kessler 6 scale among hospital nurses in Vietnam

PLOS ONE

Dear Dr. Kawakami,

Thank you for submitting your manuscript to PLOS ONE. After careful consideration, we feel that it has merit but does not fully meet PLOS ONE’s publication criteria as it currently stands. Therefore, we invite you to submit a revised version of the manuscript that addresses the points raised during the review process.

===In order to provide a more complete information to our readers on the topic, we would like to emphasize the importance to cross referencing very recent material on the same topic published in "PLoS ONE ". Therefore, it would be highly appreciated if you would check the contents published in the last two years of "PLoS ONE" (https://journals.plos.org/plosone/) and add all material relevant to your article to the reference list.

We would appreciate receiving your revised manuscript by Apr 24 2020 11:59PM. To enhance the reproducibility of your results, we recommend that if applicable you deposit your laboratory protocols in protocols.io, where a protocol can be assigned its own identifier (DOI) such that it can be cited independently in the future. For instructions see: http://journals.plos.org/plosone/s/submission-guidelines#loc-laboratory-protocols

We look forward to receiving your revised manuscript.

Kind regards,

Wen-Jun Tu

Academic Editor

PLOS ONE

Journal Requirements:

"NK reports grants from Infocom Corp, Fujitsu Ltd, Fujitsu Software Technologies, and TAK Ltd, personal fees from Occupational Health Foundation, Japan Dental Association, Sekisui Chemicals, Junpukai Health Care Center, Osaka Chamber of Commerce and Industry, outside the submitted work. The other authors declare that they have no competing interests."

Reviewers' comments:

Reviewer's Responses to Questions

**Comments to the Author**

1. Is the manuscript technically sound, and do the data support the conclusions?

Reviewer #1: Partly

2. Has the statistical analysis been performed appropriately and rigorously? 

Reviewer #1: Yes

3. Have the authors made all data underlying the findings in their manuscript fully available?

Reviewer #1: No

4. Is the manuscript presented in an intelligible fashion and written in standard English?

Reviewer #1: Yes

5. Review Comments to the Author

Reviewer #1: The report presents the findings from a study on the reliability and validity of the Kessler 6 scale among nurses in Vietnam. The study includes relatively large sample size for validation study and used appropriate statistical methods. However, there are some issues that need to be addressed.

1. The reference no. 2 "Kessler RC et al, 2003" was conducted with the aim of screening for Serious Mental Illness. Hence, the cut-off for estimated prevalence is high i.e. 13. It is not clear why the authors simply used the same cut-off for the detection of psychological distress. This cut-off will essentially underestimate the magnitude of mental health conditions in the population.

2. The scoring explanation given in lines 107 - 109, "5=all of the time"... contradicts with the information in the following sentence. Please make it clear how the responses were scored.

3. Lines 127 - 128, the pilot study was done in 150 nurses who also participated in the subsequent validation study. This is not methodologically correct. Those 150 nurses' response could be affected by their prior experience with the questionnaire.

4. If the aim is to use the K6 in community samples to screen for psychological distress, the validation conducted in health professionals is less informative. Nurses are more likely to be familiar with questions related to psychological symptoms than the general population.

6. PLOS authors have the option to publish the peer review history of their article (what does this mean?). If published, this will include your full peer review and any attached files.

Reviewer #1: Yes: Markos Tesfaye

---

## [Author Response · Author response to Decision Letter 0]

21 Mar 2020

Response to Reviewers (PONE-D-19-34231)

To Dr Wen-Jun Tu, Academic Editor

Thank you very much for your positive evaluation and valuable comments/suggestions from a reviewer for our manuscript submitted to Plos One, titled ‘Internal consistency reliability, construct validity, and item response characteristics of the Kessler 6 scale among hospital nurses in Vietnam’ (PONE-D-19-34231). We carefully studied the comments from you (the academic editor) and the reviewer and prepared a “Response to Reviewers” letter responding to these, together with a marked up and unmarked copies.

1. Following your recommendation, we add one article published in the last two years of "PLoS ONE", that is relevant to our study:

van Heyningen T, Honikman S, Tomlinson M, Field S, Myer L. Comparison of mental health screening tools for detecting antenatal depression and anxiety disorders in South African women. PLoS One. 2018;13(4):e0193697. doi: 10.1371/journal.pone.0193697.

2. We deposit our data at figshare.com: https://doi.org/10.6084/m9.figshare.12005511.v1

Please kindly update our Data Availability Statement.

3. Based on your recommendation, we studied “protocols.io” to register our laboratory protocols, However, we found that this is not very relevant to our study. Thus we do not register our protocol. We hope that this is acceptable.

To Dr. Markos Tesfaye, the reviewer #1

Reviewer #1: The report presents the findings from a study on the reliability and validity of the Kessler 6 scale among nurses in Vietnam. The study includes relatively large sample size for validation study and used appropriate statistical methods. However, there are some issues that need to be addressed.

RESPONSE: Thank you for your valuable time and effort for reviewing on our manuscript and giving us valuable comments. The below is our responses to the comments. Please also see revisions in the manuscript that are highlighted in yellow color in the marked-up copy in a separate file.

1. The reference no. 2 "Kessler RC et al, 2003" was conducted with the aim of screening for Serious Mental Illness. Hence, the cut-off for estimated prevalence is high i.e. 13. It is not clear why the authors simply used the same cut-off for the detection of psychological distress. This cut-off will essentially underestimate the magnitude of mental health conditions in the population.

RESPONSE: We agree that using the cut off score of 13+ proposed by Kessler et al., 2003 may underestimate psychological distress. We deleted the presentation of the proportions of respondents with K6 score of 13+ from Table 2, and related descriptions from the Method and Discussion. Instead, we add an additional analysis of screening performance of K6 predicting cases with common mental disorders defined by the cut-off scores of DASS21. The calculated best cut-off score of K6 was 9+, with 129 (12.4%) of the participants meeting the criteria. This would provide readers with more insights on the prevalence of psychological distress in this population. We added the related descriptions in the Methods, Results, and Discussion.

“A screening performance of K6 for identifying a CMD (i.e., depressive disorders and anxiety disorders) was also examined. Cases with CMD were defined as those who had 34 or greater scores of the total DASS21, that was the best cut-off score found in a previous study in Vietnam [34]. An area under the curve (AUC) of a receiver operative curve (ROC) and the 95% CIs were calculated predicting these cases with CMD on the sore of K6. An optimum cut-off score was determined based on the best Youden index (a sum of sensitivity and specificity), and the sensitivity, specificity, and positive likelihood ratios were calculated with their standard errors [44].” (page 8-9, Methods)

“Among the participants, 22 cases were identified as having CMD (i.e., 34 or greater scores of the total DASS21). The AUC predicting CMD on K6 score was 0.919 (95% CIs, 0.855-0.984). The best cut-off score for K6 was identified as 9+, with 129 (12.4%) of the participants having the scores equal to or greater than this cut-off. The sensitivity and specificity were 0.864 (SE, 0.073) and 0.876 (SE, 0.031), respectively, with the positive likelihood ratio of 6.96 (SE, 0.121).” (page 13, Results)

“The screening performance of K6 score predicting CMD defined by using DASS21 was high: the AUC was 0.919, with the sensitivity of 0.864 and the specificity of 0.876 for the best cut off score of 9+. The finding is similar to one from a previous study in Japan that the stratum-specific likelihood ratio became higher (>11) for the cut off scores of 9+ on K6 [46]. However, the screening performance and the best cut off score of the Vietnamese version of the K6 should be assessed by comparing people who have or have not been previously diagnosed with mental disorders.” (page 18, Discussion)

2. The scoring explanation given in lines 107 - 109, "5=all of the time"... contradicts with the information in the following sentence. Please make it clear how the responses were scored.

RESPONSE: We are very sorry that the description was wrong. We corrected this as follows sticking to the original scoring method.

“(4=All of the time, 3=Most of the time, 2=Some of the time, 1=A little of the time, and 0=None of the time)” (page 5, Methods – Measures)

3. Lines 127 - 128, the pilot study was done in 150 nurses who also participated in the subsequent validation study. This is not methodologically correct. Those 150 nurses' response could be affected by their prior experience with the questionnaire.

RESPONSE: This is a quite important point and a large limitation of our study. We added a description on this problem in the Discussion:

“Finally, 150 nurses who responded to a pilot study were also invited to the main validation study; most of them probably actually participated. Because the pilot study was anonymous, it was impossible to conduct an analysis excluding those who participated in the two surveys. Repeated surveys have been reported to result in attenuated responses to anxiety symptoms of respondents [48], which may affect the finding of the present study.” (page 18, Discussion)

4. If the aim is to use the K6 in community samples to screen for psychological distress, the validation conducted in health professionals is less informative. Nurses are more likely to be familiar with questions related to psychological symptoms than the general population.

RESPONSE: We primarily aimed to validate the K6 scale to screen psychological distress among nurses in Vietnam. However, it should be noted that the validation of the current Vietnamese version of K6 need to be done in other occupational groups or the community population if the scale is to be applied to these groups/populations. We expanded our limitation description to make this clear to readers.

“First, we aimed to validate the K6 scale to screen psychological distress among nurses, a special occupational group under high stress, in Vietnam. Nurses are usually more educated than general workers, and also familiar with questions of psychological symptoms included in K6. The present findings may not be generalized to the general population or working population in Vietnam. The reliability and validity of the present scale should be further tested in these populations.” (page 18, Discussion)

---

## [Decision Letter · Decision Letter 1]

6 Apr 2020

PONE-D-19-34231R1

Internal consistency reliability, construct validity, and item response characteristics of the Kessler 6 scale among hospital nurses in Vietnam

PLOS ONE

Dear Dr. Kawakami,

Thank you for submitting your manuscript to PLOS ONE. After careful consideration, we feel that it has merit but does not fully meet PLOS ONE’s publication criteria as it currently stands. Therefore, we invite you to submit a revised version of the manuscript that addresses the points raised during the review process.

In order to provide a more complete information to our readers on the topic, we would like to emphasize the importance to cross referencing very recent material on the same topic published in "PLoS ONE ". Therefore, it would be highly appreciated if you would check the contents published in the last two years of "PLoS ONE" (https://journals.plos.org/plosone/) and add all material relevant to your article to the reference list.

We would appreciate receiving your revised manuscript by May 21 2020 11:59PM. To enhance the reproducibility of your results, we recommend that if applicable you deposit your laboratory protocols in protocols.io, where a protocol can be assigned its own identifier (DOI) such that it can be cited independently in the future. For instructions see: http://journals.plos.org/plosone/s/submission-guidelines#loc-laboratory-protocols

We look forward to receiving your revised manuscript.

Kind regards,

Wen-Jun Tu

Academic Editor

PLOS ONE

Reviewers' comments:

Reviewer's Responses to Questions

**Comments to the Author**

1. If the authors have adequately addressed your comments raised in a previous round of review and you feel that this manuscript is now acceptable for publication, you may indicate that here to bypass the “Comments to the Author” section, enter your conflict of interest statement in the “Confidential to Editor” section, and submit your "Accept" recommendation.

Reviewer #1: All comments have been addressed

2. Is the manuscript technically sound, and do the data support the conclusions?

Reviewer #1: Yes

3. Has the statistical analysis been performed appropriately and rigorously? 

Reviewer #1: Yes

4. Have the authors made all data underlying the findings in their manuscript fully available?

Reviewer #1: Yes

5. Is the manuscript presented in an intelligible fashion and written in standard English?

Reviewer #1: Yes

6. Review Comments to the Author

Reviewer #1: The comments on the previous version have been satisfactorily addressed. It is good that the limitations are now elaborated in the discussion.

7. PLOS authors have the option to publish the peer review history of their article (what does this mean?). If published, this will include your full peer review and any attached files.

Reviewer #1: Yes: Markos Tesfaye

---

## [Author Response · Author response to Decision Letter 1]

27 Apr 2020

To Dr Wen-Jun Tu, Academic Editor

Thank you very much for your positive evaluation of our manuscript submitted to Plos One, titled ‘Internal consistency reliability, construct validity, and item response characteristics of the Kessler 6 scale among hospital nurses in Vietnam’ (PONE-D-19-34231R1). Following your recommendation, we add one more article published in "PLoS ONE", that is relevant to our study:

Choi SK, Boyle E, Burchell AN, Gardner S, Collins E, Grootendorst P, et al. Validation of six short and ultra-short screening instruments for depression for people living with HIV in Ontario: results from the Ontario HIV Treatment Network Cohort Study. PLoS One. 2015;10(11). doi: https://doi.org/10.1371/journal.pone.0142706.

We hope that the manuscript is now acceptable for the publication in Plos One.

Thank you.

Norito Kawakami, MD

On behalf of the authors

---

## [Editor Report · Decision Letter 2]

29 Apr 2020

Internal consistency reliability, construct validity, and item response characteristics of the Kessler 6 scale among hospital nurses in Vietnam

PONE-D-19-34231R2

Dear Dr. Kawakami,

We are pleased to inform you that your manuscript has been judged scientifically suitable for publication and will be formally accepted for publication once it complies with all outstanding technical requirements.

With kind regards,

Wen-Jun Tu

Academic Editor

PLOS ONE
---

## [Editor Report · Acceptance letter]

7 May 2020

PONE-D-19-34231R2 

Internal consistency reliability, construct validity, and item response characteristics of the Kessler 6 scale among hospital nurses in Vietnam 

Dear Dr. Kawakami:

I am pleased to inform you that your manuscript has been deemed suitable for publication in PLOS ONE. Congratulations! Your manuscript is now with our production department. 

With kind regards,

on behalf of

Dr. Wen-Jun Tu 

Academic Editor

PLOS ONE